# Multi-Focus Images Fusion for Fluorescence Imaging Based on Local Maximum Luminosity and Intensity Variance

**DOI:** 10.3390/s24154909

**Published:** 2024-07-29

**Authors:** Hao Cheng, Kaijie Wu, Chaochen Gu, Dingrui Ma

**Affiliations:** Department of Automation, Shanghai Jiao Tong University, Dongchuan Road 800, Shanghai 200240, China; jiaodachenghao@sjtu.edu.cn (H.C.); kaijiewu@sjtu.edu.cn (K.W.); martinbro@sjtu.edu.cn (D.M.)

**Keywords:** fluorescence imaging, multi-focus images fusion, 3D surface model

## Abstract

Due to the limitations on the depth of field of high-resolution fluorescence microscope, it is difficult to obtain an image with all objects in focus. The existing image fusion methods suffer from blocking effects or out-of-focus fluorescence. The proposed multi-focus image fusion method based on local maximum luminosity, intensity variance and the information filling method can reconstruct the all-in-focus image. Moreover, the depth of tissue’s surface can be estimated to reconstruct the 3D surface model.

## 1. Introduction

Obtaining an all-in-focus image is very important for the automation of fluorescence image collection. However, due to the limited depth of field of high-resolution fluorescence microscope, it is hard to capture an image with all objects in focus, especially for thick tissue slices [1,2].

The confocal microscope was proposed and built to adjust the stage or illumination spot to find a focus position for every pixel. These pixels can then be combined to obtain an image with all parts in focus [3,4,5]. Depth information can also be used to reconstruct the 3D model of the visible area [6,7,8]. However, it is too time-consuming to use this technique for obtaining high-resolution images.

On the other side, a simpler type of fluorescence microscope called epifluorescence microscope can obtain one image at a time, but many parts of tissue may be out of focus. So, it is reasonable to obtain images at different depths and combine those images together to obtain an all-in-focus image [9].

Normal image fusion methods have been proposed by many people. Wang et al. proposed the complex shearlet features-motivated generative adversarial network. With of help of the generative adversarial network, the whole procedure of multi-focus fusion is modeled to be the process of adversarial learning [10]. Dong et al. proposed a new image fusion framework by utilizing area-based standard deviation in the dual tree contourlet transform domain [11]. De et al. proposed a method based on wavelet transformation and maximum sharpness [12]. Li et al. proposed a method based on guided filter [13]. Guo et al. proposed a method based on self-similarity and defocus information provided by a method given by Zhuo and Sim [14,15]. The latter two methods both assign a weighted average of pixels from different source images to the corresponding pixel in the fused image. The final images fused by these methods are not clear in some cases and it is hard to obtain the depth of tissue for every pixel as in a confocal microscope.

Actually, it is reasonable to make use of the property of fluorescence images that the luminosity intensity and clarity have a strong correlation due to the architecture of fluorescence microscopes [16]. This property has also been utilized to determine the depth of a pixel in confocal fluorescent microscopes. So, we introduce the local maximum luminosity into the sharpness evaluation function. In our method, we combine it with the variance of intensity that is usually used to measure the sharpness of common images to form a new sharpness evaluation function suitable for fluorescence images [17,18].

However, unlike a confocal microscope, which has a pinhole at the detector to block out-of-focus fluorescence, epifluorescence microscopes will receive out-of-focus fluorescence which will contaminate other portions of collected images. This kind of contamination also interferes with the methods using weighted averages in the works of Li et al. and Guo et al. [13,14].

As a result of out-of-focus fluorescence, the method used for confocal microscopes does not work well for images collected using epifluorescence microscopes [19]. In order to prevent out-of-focus fluorescence, we divide the whole visible area into different rectangular blocks and use the new sharpness evaluation function we propose as a criterion of clarity. Additionally, the depth of tissue’s surface in each block can be estimated according to the criterion. Meanwhile, in order to alleviate the blocking effects introduced during block segmentation, we propose an information filling method.

According to the aforementioned ideas, a block-based image fusion method for multi-focus fluorescence microscopic images is proposed. The depth information of the tissue’s surface for each pixel can be used to reconstruct the 3D surface of observed objects when we know the stride of the image collection and parameters of the fluorescence microscope. Our contributions are presented as follows:A block-based image fusion method for multi-focus fluorescent imaging is proposed, and it is based on the local maximum luminosity, variance of intensity, and an information filling method. This method benefits from the architecture of fluorescence microscopy.A method of information filling for neighboring blocks is proposed to deal with the blocking effect introduced by the common block-based method.The depth information of each pixel can be obtained, and it can be used to reconstruct the 3D surface of these objects within source images.

The paper is organized as follows: Our methods are introduced in detail in Section 2. The comparison of images obtained by the proposed method and other methods is presented in Section 3. The discussion and conclusion are separately shown in Section 4 and Section 5.

## 2. Our Method

The proposed method is a block-based image fusion method which is mainly based on local maximum luminosity, variance of intensity within blocks, and the information filling method. The proposed method gives an explicit index per pixel which contains the depth information, allowing the feasible reconstruction of a 3D surface of the observed objects. The framework of our method can be viewed in Figure 1.

Figure 1 shows that in the first step, we collect images of the same visible area at different focal planes. Then, the whole visible area is divided into nonoverlapping rectangular blocks for block segmentation. After that, the depth can be estimated for every block, and a confidence map can be established. Confidence maps are used to adjust the depth map to further prevent the contamination of out-of-focus fluorescence. In the information filling step, an information filling method is used to alleviate blocking effects. Next, smoothing is applied to the block edges in depth space and the fused image. If the objects in the visible area are not separable, this fused image will be the final result. Otherwise, we will detect the objects and determine the segmentation block sizes according to the scale of objects. For every segmentation block size, we re-divide the whole visible area into blocks and fuse the source images again to form a new fused image. On the other side, we use the results of object detection to form masks to extract the objects from the corresponding fused images. Finally, we combine the results of different objects to obtain the final fused image.

Next, we will give more detailed information about the five key steps in our proposed method including image collection and block segmentation, the rough construction of the depth map according to the proposed sharpness estimation function and the construction of confidence map, depth adjustment according to the confidence map, pixel refinement using the information filling algorithm, and an optional step to decide the block size according to an object’s scale.

### 2.1. Image Collection and Block Segmentation

The first step is to collect source images at different focal planes and divide the whole visible area into nonoverlapping blocks. We use an automatic fluorescence scanner developed by Powerscin when collecting the data.When the stage does not move horizontally, the area we can see in the microscope is called the visible area for the sake of simplicity. In order to make sure every part of the visible area is clear in at least one source image, the range within which we collect images is between the focal planes of the deepest and shallowest part of a tissue slice in the visible area. Images are collected at different focal planes with a step size less than the microscope’s depth of field. Here, we assume *N* source fluorescence images are obtained, and these images are denoted by $I_{i}$. The whole visible area is divided into nonoverlapping rectangular blocks with size *m* × *n* (denoted by $B_{j,k}$). The set of pixels in an image $I_{i}$ within a block $B_{j,k}$ is called a patch in this paper (denoted by $p_{i,j,k}$).

Here, we decide to first estimate the depth of every block and then obtain the depth and value of every pixel within this block rather than trying to obtain the value of every pixel directly. The reason is that we assume the depth of the tissue’s surface does not change seriously within a block in most cases if the block size is not large, and actually estimating sharpness for a block is much faster than estimating the sharpness for every pixel in the block with a receptive field of the same size. If we want to estimate sharpness for every pixel faster and take a smaller receptive field size, out-of-focus fluorescence from other parts of the tissue will probably interfere with the sharpness estimation of parts that themselves are not luminous.

### 2.2. Rough Construction of Depth Map

The second step gives a rough estimation of depth for every block $B_{j,k}$ and builds up a confidence map $C_{j,k}$. In this subsection, we first introduce how to establish the depth map and show why we choose the multiplication of local maximum luminosity and variance of intensity as our sharpness evaluation function in this step. Then, we will explain how the confidence map is established.

In this step, we use the sharpness evaluation function to estimate the sharpness of every patch $p_{i,j,k}$, and the sharpness value is $s_{i,j,k}$ respectively. Rather than estimating the depth for every block $B_{j,k}$ directly, we can first determine in which image block $B_{j,k}$ is the clearest. Because we can know the object distances at which these images are taken, it is easy to convert the index of image to the depth. The index of image in which the block $B_{j,k}$ is the clearest can be represented by
(1)$${\mathrm{index}}_{j,k}=\underset{i}{\arg\max}s_{i,j,k}$$

Meanwhile, the depth information of every block $B_{j,k}$ can be inferred from $index_{j,k}$.

Because of the architecture of fluorescence microscopes, the luminosity intensity and clarity have a strong correlation, and in order to make use of this property, we introduce the local maximum luminosity into the sharpness estimation function. On the other side, the variance of intensity in a block itself can be a criterion to estimate sharpness and has good performance [17,18]. So, to take both luminosity and unevenness into consideration, we choose the multiplication of local maximum luminosity and variance of intensity as the sharpness estimation function, and it can be represented by
(2)$$s_{i,j,k}=m_{i,j,k}\times {\mathrm{var}}_{i,j,k}$$
where $m_{i,j,k}$ is the maximum pixel value in patch $p_{i,j,k}$ and
(3)$${\mathrm{var}}_{i,j,k}=\frac{1}{mn}\sum_{v_{r,s}\in p_{i,j,k}}{\left(v_{r,s}-\mu_{i,j,k}\right)}^{2}$$

Here, $\mu_{i,j,k}$ is the average flux for every patch $p_{i,j,k}$, and $v_{r,s}$ is the pixel value in the corresponding patch $p_{i,j,k}$. This criterion is better than other current sharpness evaluation functions for fluorescent images. This is better than the method which selects the maximum pixel value for every pixel which is used for confocal microscopes.

In order to show the performance, we compare our proposed sharpness evaluation function with other sharpness evaluation functions. The visible area is divided into blocks with size 20 × 20. For every sharpness evaluation function, we select a patch $patch_{j,k}$ for every block $B_{j,k}$ and
(4)$${\mathrm{patch}}_{j,k}=p_{{\mathrm{index}}_{j,k},j,k}$$

Then, we stitch patches together to obtain a fused image for each sharpness evaluation function. Additionally, we show the result of the method used for confocal microscopes which selects the maximum pixel value for every pixel. We show the comparison in Figure 2. The images in the left column of Figure 2 show the whole fused images, and the images in the center and right columns of Figure 2 show two typical regions extracted from the whole visible area.

Figure 2a–c give the performance of maximum total luminosity method to extract the brightest values for every pixel, and the method is used in confocal microscopes [3,4]. We can see the texture contaminated by the out-of-focus fluorescence. The following images are reconstructed by selecting the patches $patch_{j,k}$ for the corresponding sharpness evaluation functions.

Figure 2d–f use the Laplacian measure which is used by Zhuo and Sim as the criterion of clarity. The textures are not clear due to the contamination [15].

Figure 2g–i use the variance measure as a typical criterion of clarity [17,18]. This method has good performance for the most part, but it is not robust and the reconstructed image lacks continuity.

Figure 2j–l use the normalized-variance measure as the criterion of clarity [17,18]. It is even less robust than the variance measure.

Figure 2m–o use local maximum luminosity as the criterion of clarity. It obtains good performance for the most part and is robust. It also shows maximum continuity and good performance, but the textures in some parts are less clear than those of the variance measure.

Figure 2p–r use the proposed multiplication of the local maximum luminosity and intensity variance in the block as the criterion of clarity. It shows good performance and is very robust.

It is obvious that the performance of the multiplication of local maximum luminosity and variance of intensity is the best sharpness evaluation function for fluorescent images. The method used for the confocal microscope fails because of contamination of out-of-focus fluorescence. The Laplacian kernel is applied in the method proposed by Li et al., and it does not give good performance [13]. Compared with variance and normalized variance, it improves the robustness and continuity. Compared with local maximum luminosity, it is more likely to find the patch with a clearer texture. So, the multiplication of the local maximum luminosity and variance of intensity is selected as the sharpness evaluation for the second step of our method.

After that, we can establish the confidence map. The fluorescent substance will emit more light when it is in focus than when it is not. The depth estimation of the tissue parts that are luminous enough when in focus is less vulnerable to out-of-focus fluorescence from other parts. Based on this assumption, the confidence map has the following form:(5)$$C_{j,k}=\left\{\begin{array}{cc} 0 & m_{j,k}\leq \;{\mathrm{threshold}}_{0}\\ 1 & m_{j,k}>\;{\mathrm{threshold}}_{0} \end{array}\right.$$
where ${threshold}_{0}$ here is a parameter that can be adjusted, and $m_{j,k}$ is the maximum pixel value among $patch_{j,k}$.

### 2.3. Depth Adjustment

In the third step, in order to prevent the contamination of out-of-focus fluorescence in the void regions, information in the confidence map $C_{j,k}$ is used to adjust the depth for each block. Because the surface of the tissue is continuous, the estimated depths of adjacent blocks should not differ a lot in most cases. During this step, if the depth of a block has a large difference from those of the surrounding blocks, the depth will be adjusted according to Algorithm 1. In this manner, lots of contaminated parts will be recovered.
**Algorithm 1** Adjust $index_{j,k}$1:vote_num←02:pos_vote←03:**for** sj=j−kernel:j+kernel **do**4:    **for** sk=k−kernel:k+kernel **do**5:       **if** $c_{sj,sk}==1$ **then**6:          vote_num←vote_num+17:          **if** $|index_{j,k}-index_{sj,sk}|>threshold_{1}$ **then**8:             pos_vote←pos_vote+19:          **end if**10:      **end if**11:   **end for**12:**end for**13:**if** $pos\_vote>ratio\times vote\_num\;\;\mathrm{AND}\;\;vote\_num>threshold_{2}$ **then**14:   $index_{j,k}\leftarrow {\mathrm{Average}}_{sj,sk}\;index_{sj,sk}$15:**end if**


For every block $B_{j,k}$, we adjust the $index_{j,k}$ with the following Algorithm 1.

Here, the kernel, ${threshold}_{1}$, ratio, and ${threshold}_{2}$ are independent parameters which can be adjusted. After that, for every block where $C_{j,k}=0$ and there is a neighboring block that is confident, we set it to the average of $index_{sj,sk}$ where (sj,sk) near (j,k) and $C_{sj,sk}=1$.

### 2.4. Information Filling

The blocking effects introduced by block segmentation can be obvious with the assumption that the depths of pixels within a block that are similar are invalid.The boundaries of objects are typical examples, and in this situation, the depths of some pixels within the block may be far away from the estimated depth of the block but close to the estimated depths of adjacent blocks. Therefore, in the fourth step, a method of information filling is used to alleviate the blocking effects and assign a depth to every pixel. After this step, every pixel has its own depth, and this information is useful for the 3D modeling of objects in the visible area. First, we initialize the index for every pixel ${pixindex}_{pj,pk}=index_{j,k}$ where pixel $pix_{pj,pk}$ belongs to $B_{j,k}$. The part of this algorithm is given in Algorithm 2 for any two adjacent block $B_{j1,k1}$, $B_{j2,k2}$ where at least one of their confidence values is 1.

**Algorithm 2** Information filling.
1:**if** $|index_{j1,k1}-index_{j2,k2}|>threshold_{3}$ **then**2:   **for** every pixel $pix_{pj,pk}$ belongs to $B_{j1,k1}$ or $B_{j2,k2}$ **do**3:     The corresponding pixel value in every source image is $v_{i,pj,pk}$4:     **if** $p_{pj,pk}$ belongs to $B_{j1,k1}$ **then**5:        $d=pixindex_{pj,pk}$; $x=index_{j2,k2}$6:        **if** $v_{d,pj,pk}<threshold_{4}$ AND $v_{d,pj,pk}<v_{x,pj,pk}$ **then**7:          $pixindex_{pj,pk}=x$8:        **end if**9:     **else**10:        $d=pixindex_{pj,pk}$; $x=index_{j1,k1}$11:        **if** $v_{d,pj,pk}<threshold_{4}$ AND $v_{d,pj,pk}<v_{x,pj,pk}$ **then**12:          $pixindex_{pj,pk}=x$13:        **end if**14:     **end if**15:   **end for**16:
**end if**



Here, ${threshold}_{3}$ and $threshold_{4}$ are two independent parameters which can be adjusted. After that, the smoothing is applied to the edge of each block in depth space and fused image. After this step, we can obtain a useful depth map.

### 2.5. Optional Step

Optional steps can be applied for objects that can be separated (an example can be seen in Figure 3); after, segmentation methods like DBSCAN or group finding methods like FoF can be applied to separate different objects [20,21].

To take suitable segmentation block sizes for different objects, the block sizes used to re-divide the visible area will be adjusted according to the scale of these objects. Objects of the same block size can be merged into a class to be processed together, and this will take less time. Then, pixels that do not belong to any class are assigned to the nearest classes, but one pixel that has been assigned to one class will not be assigned to other classes. The domains occupied by one class can be treated as a mask (the pixel of source images will be treated as 0 if the pixel is outside of this mask) when processing this class. The steps aforementioned will be used to fuse the images for every class.

After all the classes of objects are fused in the manner of algorithms mentioned before, they can be combined together to obtain the final fused image. The depth maps obtained for different classes of objects will be combined with the help of these masks. The flow chart can also be seen in Figure 1. This step has no effect on tissue that is not separable, so for this kind of tissue, we do not need to take this step.

## 3. Experiments

The source images are obtained using the fluorescence microscope at different object distances. All experiments are run on a laptop with 4 Cores, 2.6 GHz CPU, and 16 GB RAM. Here, the proposed method is compared with a method proposed by De et al., GFF (guided filtering-based fusion method), SSS (shared self-similarity) and SSSDI (shared self-similarity and depth information) [12,13,14].

The method proposed by De et al. makes use of the wavelet transformation [12]. In their method, they first use a nonlinear wavelet to decompose the source images into multi-resolution signals. Then, fusion happens at different resolutions to pursue maximum sharpness. Finally, they reconstruct the fused image through composition. GFF decomposes images into base layers and detail layers and uses a weighted average method for fusion separately for base layers and detail layers [13]. The weight maps are constructed with the help of guided filtering. The final fused image is obtained by combining the fused base layer and detail layer. SSS and SSSDI were both proposed by Guo et al., and they are very similar [14]. SSS is also a weighted average method. It makes use of shared self-similarity to generate adaptive regions and choose SML (some-of-modified-Laplacian) as a clarity metric. The fusion weights depend on this clarity metric. Different from SSS, SSSDI divides the clarity metric used by SSS by the square of the defocus scale to form a new clarity metric [14]. The defocus scale can be estimated by
(6)$$\sigma \left(x,y\right)=\frac{1}{\sqrt{R{\left(x,y\right)}^{2}-1}}\sigma_{0}$$
where $\sigma_{0}$ is a given fixed value, and R(x,y) can be represented by
(7)$$R\left(x,y\right)=\frac{\left|\nabla i(x,y)\right|}{\left|\nabla i_{1}\left(x,y\right)\right|}$$

Here, ∇i(x,y) is the gradient of the original image at pixel (x,y) and $\nabla i_{1}\left(x,y\right)$ is the gradient of the image at pixel (x,y) after reblurring with a Gaussian kernel [13]. However, for the void region where pixel values are 0, ∇i(x,y) and $\nabla i_{1}\left(x,y\right)$ are both 0, and it is hard to define R(x,y) and σ(x,y). So, SSSDI is invalid for some regions with no fluorescence in the source images.

The method proposed by De can fuse any number of source images. However, this method cannot make sure the lower bound of pixel value is greater than or equal to 0. The two methods are tested: (1) rescaling values for every pixel according to the lower bound and upper bound of the fused image, and (2) putting all the values below 0 to 0. The second method produces better fused images, and the images processed by the second method are selected for comparison. Meanwhile, the code provided by Li et al. for GFF and Guo et al. for SSS and SSSDI can fuse two source images at a time. In order to fuse *N* source images, images of an odd index are fused with the following images with an even index. If the last image has an odd index, it remains to the next round of fusion, and we iterate this process until the final fused image is obtained. For the proposed method, the multiplication of local maximum luminosity and variance of intensity is chosen as the $s_{i,j,k}$.

### 3.1. Fusion Performance

The performance of different methods on one set of source images is shown in Figure 3. This set of source images includes 81 fluorescence images of one kind of pollen. The parameters for GFF, SSS and SSSDI are the default parameters in their code. The parameters for our method are set as $threshold_{0}=60$, kernel=2, $threshold_{1}=40$, $threshold_{2}=4$, ratio=0.5, $threshold_{3}=10$, $threshold_{4}=80$, and the tentative block size is 15×15.

It can be seen that the method proposed by De and Chanda is not vulnerable to contamination of out-of-focus fluorescence, but it suffers from artifact-like blocking effects [14]. Additionally, it cannot make sure the lower bound is zero, and this is another problem of this method. The result of GFF suffers from out-of-focus fluorescence because of the weighted average method [13]. The texture of the pollen grains in the image fused by GFF is not as clear as that in ours, especially the top left and the bottom. SSS and SSSDI methods seem more vulnerable to this interference, and SSSDI performs no better than SSS in this situation [14]. We can see that the texture of the top left pollen grain is dramatically contaminated by the out-of-focus fluorescence.

The proposed method is less vulnerable to out-of-focus fluorescence and obtains better performance. But the stability of our method in the object’s boundary is not as good as that within the boundary because the third step cannot work very well in this situation, as it is hard to obtain enough positive votes. As we can see, the proposed method has obtained state-of-the-art performance and has better performance on some parts of the pollen grains.

In order to show the generality of the proposed method, the results of different methods on another set of source images are shown in Figure 4. This set of images contains 60 fluorescence images of a spinach stem. The parameters for our method are set as $threshold_{0}=150$, kernel=2, $threshold_{1}=40$, $threshold_{2}=4$, ratio=0.5, $threshold_{3}=10$, and $threshold_{4}=80$, and the tentative block size is 20×20.

Even though most textures in the image fused by the method proposed by De are clear, it still suffers from the blocking effect [12]. SSS and SSSDI methods are still affected by the contamination of out-of-focus fluorescence if we focus on the bottom right part [14]. The fused image using the GFF method is much better than SSS and SSSDI but not as clear as that of our method [13]. What is more, our method is even faster than the other three methods in this situation [12,13,14]. We do not show the image fused by our proposed method with this optional step because this tissue should be treated as a whole, and the result will be almost the same.

### 3.2. Computation Time

The computational times of all the methods spent on the different sets of source images are shown in Table 1.

As can be seen, the proposed method without the optional step is the fastest, and the proposed with the optional step will be much slower. However, the optional step is not necessary in many situations. The method proposed by De and Chanda and the method proposed by Li et al. are fast but not as fast as the proposed one without the optional step [12,13]. The SSSDI method proposed by Guo et al. is the slowest [14]. SSSDI is claimed to take about five times as long as SSS in their paper [14]. Actually, SSSDI takes more than eight times as long as SSS in our experiments, and this is because the laptop runs out of memory RAM, and it needs to make use of the disk.

### 3.3. Depth Map

As aforementioned, the proposed method can obtain a determined depth map which can be used to reconstruct the 3D surface of observed objects if the parameters of the fluorescent are known. Here, an example of a depth map and point cloud of one pollen grain’s surface are shown in Figure 5. Figure 5a is the fused image of the first set of fluorescence images. Figure 5b is the depth map constructed in our algorithms, and the depth is set to 0 for the void region. Figure 5c is the point cloud of one pollen, which is marked by a red rectangle in Figure 5a. Better reconstruction of the 3D surface of the observed area may be achievable using the depth information obtained by our method, but it is out of the range of this paper.

## 4. Discussion

From the result of the experiment aforementioned, it can be seen that, first, our infusion method for fluorescence imaging can achieve start-of-art performance. Second, the computation time for the proposed method is on an average level, and for inseparable tissue, it is faster than the other three methods because the optional part is not necessary and the computation time will decrease. Third, the blocking effects are obviously alleviated by our information filling method. Moreover, the depth for every pixel can be obtained from the proposed method.

The proposed method can make use of the properties of epifluorescence microscopes to finish the work of a confocal microscope such as obtaining clear images and obtaining the depth information of every pixel. This information can be used to reconstruct the 3D model of the visible area similar to a confocal microscope. Compared to using other devices like a light field microscope for high-quality microscopic imaging, our method has lower equipment costs and higher algorithm efficiency. Additionally, the speed of obtaining a clear image will increase prominently. However, due to equipment limitations, we did not collect ground-truth images to conduct quantitative analysis of our algorithm. Moreover, whether our multi-depth fusion algorithm based on the sharpness evaluation function can be applied to more scenarios still requires further experiments and verification.

## 5. Conclusions

In this paper, a fluorescence image fusion method based on local maximum luminosity, variance of intensity within blocks, and the information filling method has been proposed and compared with a state-of-the-art image fusion method. In order to estimate clarity, the relationship of luminosity and clarity has been used, and a new sharpness measure has been proposed. Additionally, the performance of it has been compared with other sharpness measures. The information filling method has been proposed to alleviate blocking effects. In the experiment, it shows that we have obtained state-of-the-art performance in fusing fluorescent images, and our method is faster than the other three methods when the optional step is not necessary. What is more, the estimation of the depth can also be obtained to reconstruct the 3D model of the visible area.

## Figures and Tables

**Figure 1 sensors-24-04909-f001:**
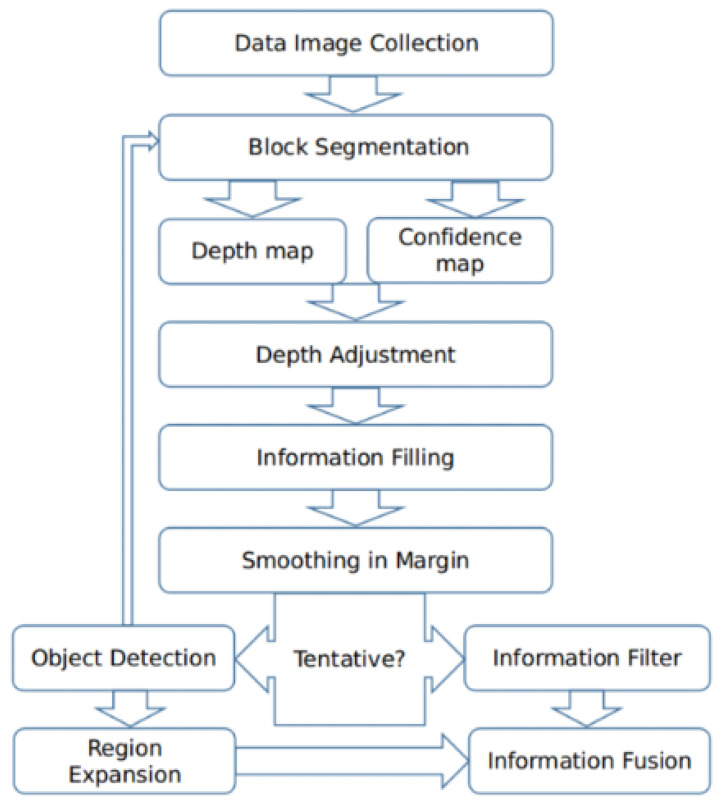
Framework of our proposed method.

**Figure 2 sensors-24-04909-f002:**
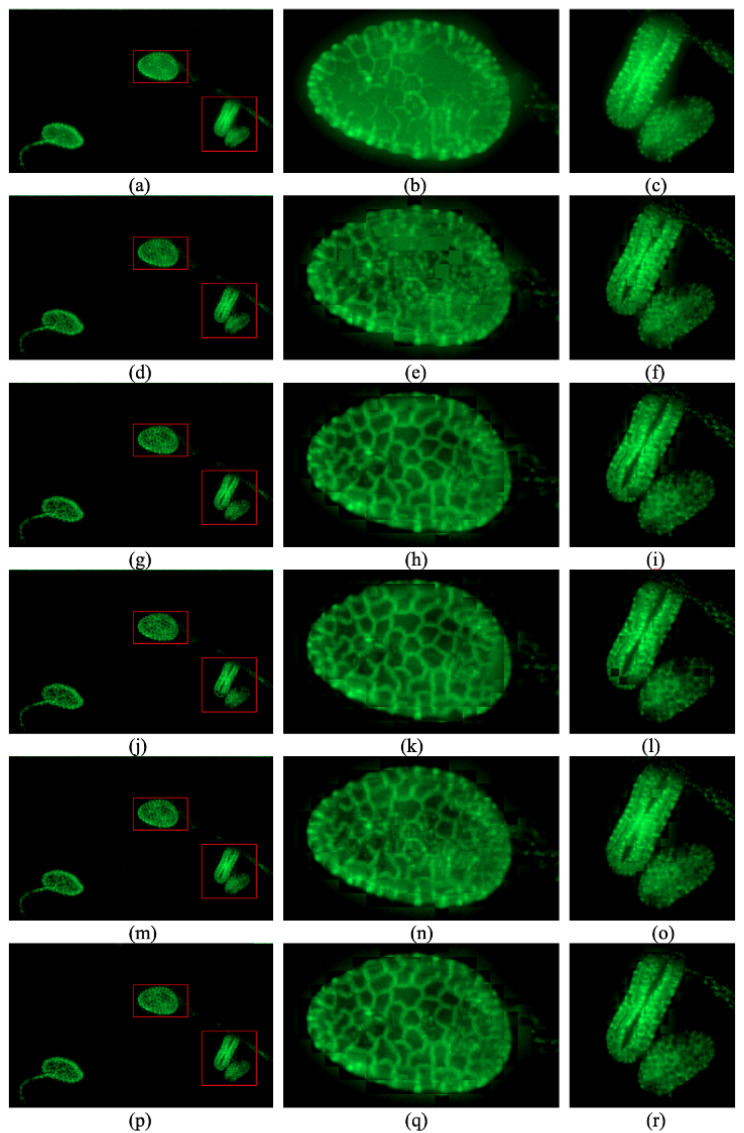
Performance of maximum total luminosity method and the results of different sharpness estimation functions: (**a**–**c**) maximum total luminosity method [3,4]. (**d**–**f**) the Laplacian measure which is used by Zhuo and Sim [15]. (**g**–**i**) a method using the variance measure as a typical criterion of clarity [17,18]. (**j**–**l**) a method using the normalized-variance measure as the criterion of clarity [17,18]. (**m**–**o**) a method using local maximum luminosity as the criterion of clarity. (**p**–**r**) our method.

**Figure 3 sensors-24-04909-f003:**
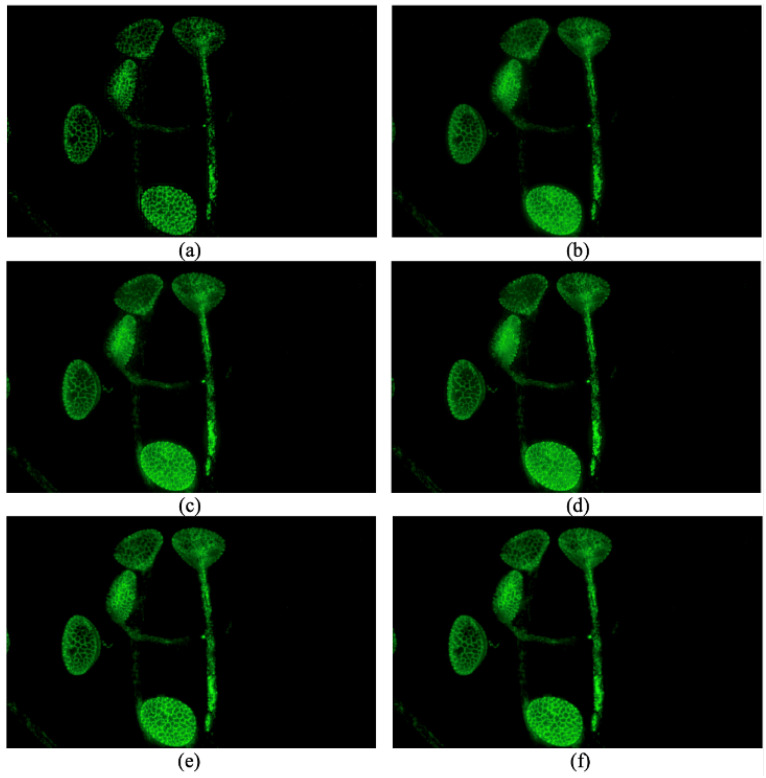
Images fused by different methods: (**a**) a method proposed by De and Chanda [12]. (**b**) GFF proposed by Li et al. [13]. (**c**) SSS proposed by Guo et al. [14] (**d**) SSSDI proposed by Guo et al. [14]. (**e**) Our proposed method without an optional step. (**f**) Our proposed method with an optional step.

**Figure 4 sensors-24-04909-f004:**
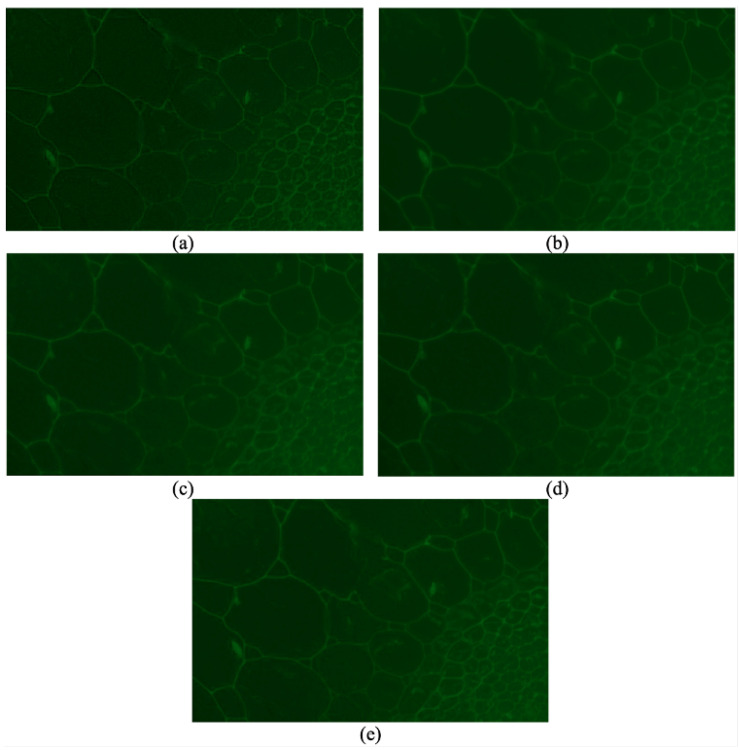
Images fused by different method: (**a**) a method proposed by De and Chanda [12]. (**b**) GFF proposed by Li et al. [13]. (**c**) SSS proposed by Guo et al. [14]. (**d**) SSSDI method proposed by Guo et al. [14]. (**e**) Our proposed method without an optional step.

**Figure 5 sensors-24-04909-f005:**
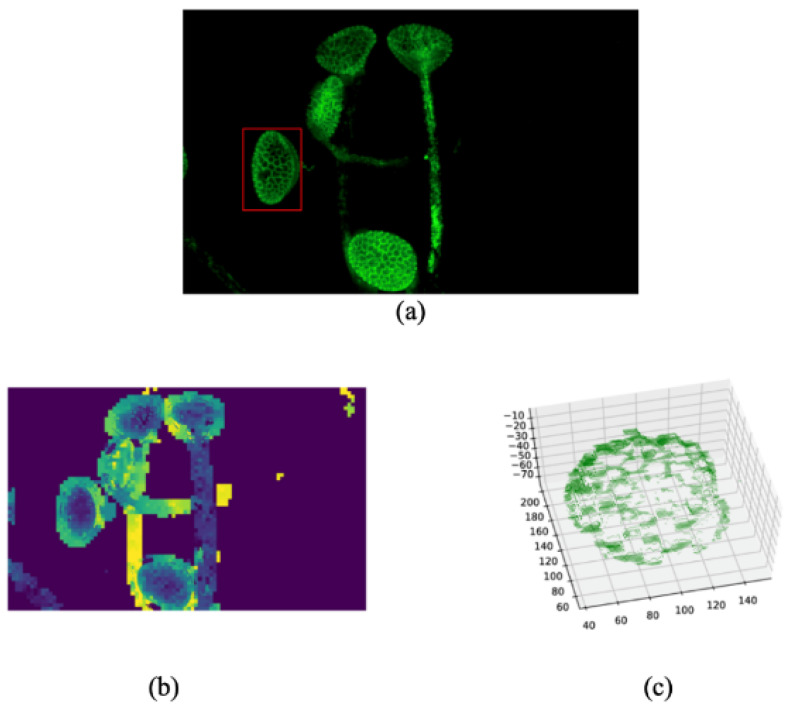
Example of depth map obtained from our method and the point cloud of one pollen: (**a**) the fused image of the first set of fluorescence images. (**b**) the depth map constructed in our algorithms. (**c**) the point cloud of one pollen.

**Table 1 sensors-24-04909-t001:** Computational time for different methods (in minutes).

	De	GFF	SSS	SSSDI	Ours	Ours (with Optional Step)
Set 1 (1920×1200)	3.48	4.38	254.67	2212.07	1.51	17.17
Set 2 (1920×1200)	2.97	3.12	111.67	1555.05	0.89	-

## Data Availability

The data presented in this study are available on request from the corresponding author.

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
