# Peer review of "Multi-Focus Images Fusion for Fluorescence Imaging Based on Local Maximum Luminosity and Intensity Variance"

_sensors, 2024, doi:10.3390/s24154909_

Round 1

Reviewer 1 Report

Comments and Suggestions for Authors

Author Response

Comments 1: In method part, I did not find any information about the data collection process.  Please describe what a microscope was utilized for data collection? Additionally, please describe the sample preparation methods. Ensuring reproducibility is important in scientific research.

Response 1: Thank you for pointing this out and we agree with this comment. Therefore, we have described the microscope utilized in our work. The data collection equipment we use is an automatic fluorescence scanner developed by Powerscin and specific parameters can be referenced on the product homepage (http://www.powerscin.com/ProDetail.aspx?ProId=144). Data collection steps have been supplemented in the text and changes can be found in line 95-102.

Comments 2: In figure 2, quantitative analysis is preferable over image inspection alone. I suggest collecting ground truth images (e.g., confocal, light sheet) and comparing them with epi-fluorescence images processed by different algorithms. Calculating SSIM for each group will provide a clearer understanding for readers.

Response 2: Thank you for pointing this out. In Figure 2, we compare the results produced by different sharpness evaluation methods. Using SSIM requires a ground truth image, but in this work, due to equipment limitations, it is difficult to obtain a clear GT image, so no quantitative analysis was conducted. However, we have conducted a detailed comparison of the results of each method. It can be intuitively seen from the images that our method has advantages in stability, continuity, and detail processing capability. Analysis can be found in line 151-169.

Comments 3: In figure 2, block artifacts are still present. Please describe this issue, and if it can't be resolved currently, include it in the discussion section.

Response 3: Figure 2 compares different sharpness evaluation algorithms. We aim to demonstrate the effectiveness of our sharpness evaluation algorithm through experiments. However, the real elimination of blocking effects occurs in the fourth step of our method, Information Filling, and the description can be found in line 196-209. Therefore, the results shown in Figure 2 still exhibit block artifacts, which is inherent to our block-based method. Of course, we will expand our Discussion, as seen in our response to Comments 5.

Comments 4: In figure 3 and 4, as previously suggested, a quantitative analysis is necessary. If obtaining images with the same FOV using different microscopes is difficult, at least provide an error map or SSIM map between the methods.

Response 4: Agree. But referring to the response to Comments 2, we have already provided a detailed analysis comparing the results of different methods, which can also be intuitively seen in the figures. We will design quantitative indicators in future research. Thank you for your suggestion.

Comments 5: While the method in this article is cost-effective and promising for advanced biomedical applications, the discussion should address its limitations. For instance, there is a lack of quantitative analysis. If such experiments are unfeasible, explain why. Additionally, include more case studies to highlight why and when this method is advantageous. Reproducibility is crucial; consider making your algorithms open source, though not mandatory. In summary, I recommend rewriting the discussion to include more limitations of the study.

Response 5: Thank you for advice and we have added the limitations. Due to equipment limitations, we did not collect GT images to conduct quantitative analysis of our algorithm. This is indeed a limitation of our research. We will supplement this in future studies. However, we believe that the qualitative analysis in this paper, combined with the quantitative indicator of computation time, can demonstrate the advantages of our algorithm. Meanwhile, we have presented the core algorithm in pseudocode form in the paper, and we welcome suggestions from colleagues. Changes can be found in line 327-333.

Comments 6: As previously mentioned, include a quantitative comparison. I suggest stabilizing the sample in gel or agarose, then acquiring the 3D distribution with a confocal or light sheet microscope. Use a widefield epifluorescence microscope for data collection and perform a quantitative analysis.

Response 6: Thank you for advice. As mentioned in the article, thanks to depth estimation, 3D reconstruction is an additional feature of our algorithm, but in-depth analysis of this exceeds the scope of this paper. This paper focuses on enhancing 2D fluorescence images, so we did not compare it with the state-of-the-art 3D reconstruction methods. Overall, our method has a significant advantage in computational efficiency and can also provide reference for 3D reconstruction. Therefore, we believe our work is meaningful.

Reviewer 2 Report

Comments and Suggestions for Authors

1. Specify image spatial resolution, depth resolution, extended depth of field, depth to resolution ratio, reconstructed cloud pixel size, FOV. Computational complexity, reconsruction speed, noise robustness. 

2. Light field imaging is a common computational imaging method to realize extended depth-of-field all-in-focus image by refocusing.

For example,  10.21203/rs.3.rs-2883279/v1 describes a light field fluorescence microscopy method to generate all-in-focus image. Can authors compare proposed methods with light field imaging methods? Discuss advantages and disadvantages

3. Can this proposed methods be applied to other imaging applications? For example, LiDAR with extended depth-of field, as realized in this paper https://www.nature.com/articles/s41467-022-31087-9

Author Response

Comments 1: Specify image spatial resolution, depth resolution, extended depth of field, depth to resolution ratio, reconstructed cloud pixel size, FOV. Computational complexity, reconstruction speed, noise robustness.

Response 1: Thank you for pointing this out and we agree with this comment. We have described the microscope utilized in our work and description can be found in line 95-102. In addition, we have conducted a quantitative analysis of the algorithm's running time. Please refer to Table 1 in the article.

Comments 2: Light field imaging is a common computational imaging method to realize extended depth-of-field all-in-focus image by refocusing. For example, 10.21203/rs.3.rs-2883279/v1 describes a light field fluorescence microscopy method to generate all-in-focus image. Can authors compare proposed methods with light field imaging methods? Discuss advantages and disadvantages.

Response 2: In this reference, the research team uses a light field microscope for high-quality 3D microscopic imaging through fluorescence lifetime imaging, which provides certain insights for our subsequent research from the perspective of equipment. Light field imaging is a revolutionary technology that records the angle and position information of light rays, providing richer visual information and stronger image processing capabilities. Although light field imaging technology has many advantages, it also faces some challenges, such as: (1) Large data volume: Capturing and processing light field data requires significant storage and computing resources; (2) High equipment cost: The cost of light field cameras and microscopes is high, limiting its widespread application; (3) Complex post-processing: Light field data processing is complex and requires specialized software and algorithm support. Our method directly processes 2D images with different depths of field, has lower equipment costs, and high algorithm efficiency, so we believe our method is meaningful. We have also added this comparative analysis to the Discussion section of the article.

Comments 3: Can this proposed methods be applied to other imaging applications? For example, LiDAR with extended depth-of field, as realized in this paper: https://www.nature.com/articles/s41467-022-31087-9

Response 3: The compact light field photography method (CLIP) proposed in this reference provides an efficient and flexible means to acquire large-scale light field data. It can achieve high-quality 3D imaging, suitable for various complex and dynamic scenes, opening new avenues for future 3D vision technology. The core of our method is to provide a sharpness evaluation algorithm based on images collected by microscopic instruments. There may be some differences when using cameras to capture images in the macroscopic world, so our method may not be directly transferable to 3D imaging in the macroscopic field. However, experiments have shown that our method has good results in fusing microscopic images at different depths of field, so we believe the research is meaningful.

Round 2

Reviewer 2 Report

Comments and Suggestions for Authors

concerns are addressed